# Fourier Ptychographic Microscopy 10 Years on: A Review

**DOI:** 10.3390/cells13040324

**Published:** 2024-02-10

**Authors:** Fannuo Xu, Zipei Wu, Chao Tan, Yizheng Liao, Zhiping Wang, Keru Chen, An Pan

**Affiliations:** 1State Key Laboratory of Transient Optics and Photonics, Xi’an Institute of Optics and Precision Mechanics, Chinese Academy of Sciences, Xi’an 710119, China; xufannuo22@mails.ucas.ac.cn (F.X.); wuzipei2020@email.szu.edu.cn (Z.W.); tanchao@stu.scu.edu.cn (C.T.); liaoyizheng22@mails.ucas.ac.cn (Y.L.); zhpwang20@lzu.edu.cn (Z.W.); chenkeru@stu.xjtu.edu.cn (K.C.); 2University of Chinese Academy of Sciences, Beijing 100049, China; 3School of Physics and Optoelectronic Engineering, Shenzhen University, Shenzhen 518060, China; 4School of Electronics and Information Engineering, Sichuan University, Chengdu 610065, China; 5School of Physical Science and Technology, Lanzhou University, Lanzhou 730000, China; 6School of Automation Science and Engineering, Xi’an Jiaotong University, Xi’an 710049, China

**Keywords:** fourier ptychographic microscopy, computational imaging, biomedical imaging

## Abstract

Fourier ptychographic microscopy (FPM) emerged as a prominent imaging technique in 2013, attracting significant interest due to its remarkable features such as precise phase retrieval, expansive field of view (FOV), and superior resolution. Over the past decade, FPM has become an essential tool in microscopy, with applications in metrology, scientific research, biomedicine, and inspection. This achievement arises from its ability to effectively address the persistent challenge of achieving a trade-off between FOV and resolution in imaging systems. It has a wide range of applications, including label-free imaging, drug screening, and digital pathology. In this comprehensive review, we present a concise overview of the fundamental principles of FPM and compare it with similar imaging techniques. In addition, we present a study on achieving colorization of restored photographs and enhancing the speed of FPM. Subsequently, we showcase several FPM applications utilizing the previously described technologies, with a specific focus on digital pathology, drug screening, and three-dimensional imaging. We thoroughly examine the benefits and challenges associated with integrating deep learning and FPM. To summarize, we express our own viewpoints on the technological progress of FPM and explore prospective avenues for its future developments.

## 1. Introduction

Microscopes serve as a critical tool for exploring the microcosm and have played an indispensable role in the advancement of human technology and civilization. Optical microscopes, in particular, achieve imaging by capturing and processing the interactions between light and the sample. Due to their non-invasive and non-contact advantages, they are widely used in fields such as biology and medicine [1,2]. With the rapid development of digital pathology, flow cytometry, and high-throughput screening technologies, the demand for high-resolution and large field-of-view (FOV) imaging is increasing, prompting continuous advancements in optical microscopy techniques [3,4,5].

Facing this challenge, a traditional solution is to increase the aperture of the objective lens to expand the FOV. However, this significantly increases manufacturing costs and complicates the control of moving and focusing the objective lens. Furthermore, this method induces geometric distortions that necessitate additional lenses for correction, making it suitable only for fixed scenarios such as steppers.

Given that the conflict between resolution and field-of-view is mostly caused by the impact of lens aberration, a potential solution is to entirely eliminate the requirement for lenses and instead utilize lensless technology. Lensless imaging technology [6,7,8,9], developed in recent years, has advantages like small size, large FOV, and low cost. This technology does not utilize lenses; instead, it employs sensors to capture the diffraction patterns generated by the object. It finds extensive applications in scenarios where lenses are difficult to manufacture, such as X-ray imaging. However, in the visible light spectrum, traditional lens-based technologies are quite mature, and microscopy techniques based on traditional microscope structures often yield better imaging results. Moreover, due to limitations in semiconductor technology and light collection efficiency, the sensor pixel size cannot be infinitely reduced, limiting the potential for resolution improvement in lensless imaging [10]. Additionally, this technology has special requirements for sample preparation, making its application range relatively narrow. Its potential for resolution improvement in the optical spectrum is relatively limited, and it struggles with phase recovery in low-frequency regions [11,12,13]. Consequently, its overall application value is constrained.

Another straightforward and efficient technique entails the use of high-magnification objectives to capture images of the sample at various points, which are subsequently combined to create a composite scan. This method guarantees a high level of detail while simultaneously increasing the FOV. Nevertheless, the stitched image frequently exhibits subpar quality, characterized by flaws such as ghosting and blurring. Moreover, the scanning procedure is characterized by a significant amount of time required, while the incorporation of a high-precision motorized stage and high-magnification targets significantly escalates the expenses associated with manufacturing. Synthetic aperture is the antithesis of standard scanning stitching, as it prioritizes achieving high resolution before obtaining a large FOV. Synthetic aperture radar technology [14,15,16] is an effective way to improve resolution in the microwave band. In optical wavelengths, there are many research studies that combine the concept with digital holography and microscopy imaging [17,18,19,20,21,22,23,24]. However, due to the much higher frequency of visible light compared to microwaves, phase measurement becomes exceptionally challenging, limiting its application in the biomedical field.

Resolving the contradiction between FOV and resolution is the sole means of restoring the intensity image. To achieve quantitative phase imaging, further technologies are required. An imaging technique that can restore the phase and obtain high-resolution images is called digital holography [25], which utilizes the principle of interference. This technique involves the diffraction calculation of the interference pattern between the object wave and the reference wave, thereby reconstructing the hologram. To enhance imaging resolution and signal-to-noise ratio, the concept of synthetic aperture was adopted, leading to the invention of synthetic aperture digital holography [26,27]. However, it is still constrained by the physical law between the FOV and the resolution, resulting in a limited spatial bandwidth product. Moreover, due to its interference-based principle, the imaging apparatus is quite complex. In addition to digital holography, techniques such as self-interference digital holography [28] and optical coherence tomography [29,30] also directly utilize light interference to measure the morphology of samples and are referred to as imaging interferometric microscopy. Another technique related to phase retrieval is called ptychography [31], which was first proposed by Hoppe et al. in 1969 as a form of coherent diffraction imaging to solve phase measurement issues in electron microscopy. This method has been extended to imaging in multiple bands over its subsequent development. In 2004, Rodenburg et al. introduced the ptychographic iterative engine (PIE) [32,33] algorithm, which achieved large FOV and high-resolution imaging by using multiple overlapping diffraction patterns in the spatial domain and recovered the phase using iterative algorithms. Maiden and Rodenburg proposed ePIE [34] in 2009, which can solve for both the sample distribution function and the probe function.

Inspired by significant advancements in optical imaging mentioned above, especially imaging interferometric microscopy (IIM), innovative use of synthetic aperture in optical bands, and ptychography’s unique iterative algorithm that alternates constraints in both the frequency and spatial domains, Zheng et al. introduced FPM [35] in 2013. This technique, deeply rooted in the principles of the aforementioned methods, blends the spatial resolution enhancement of a synthetic aperture with the computational capabilities and phase retrieval of ptychography. FPM not only offers high-resolution and expansive FOV imaging but also excels in quantitative phase imaging, gleaning crucial data from diffraction patterns. Furthermore, FPM integrates the concepts of phase recovery and synthetic aperture, marrying LED array illumination with PIE phase recovery algorithms. It sidesteps the traditional step-scanning operation in ptychography by transitioning the stitching operation to the frequency domain through multi-angle illumination [36,37]. In essence, FPM captures a series of low-resolution images under varied illumination angles, leveraging intensity information in the spatial domain and the cutoff frequency in the frequency domain as constraints. The technique then stitches these in the frequency domain while concurrently recovering the phase using iterative algorithms.

Over the past decade, FPM technology has rapidly developed. Figure 1 illustrates the evolution of related publications since its inception in 2013, with notable milestones in its development highlighted within the figure. Not only is it a means of microscopic imaging, but it has also become a computational imaging framework for dealing with optical system limitations. It has changed the traditional “what you see is what you get” imaging mode by combining the encoding of optical images with digital decoding, achieving significant improvement in imaging quality, breaking the physical limitations of optical systems, and greatly enhancing the information acquisition capability of imaging systems.

This review aims to provide a comprehensive introduction to the basic principles of FPM technology and compare it with other microscopic imaging technologies, such as structured illumination microscopy (SIM), synthetic aperture, and differential phase contrast (DPC). We also systematically summarize various applications and technological advancements based on FPM, including high-speed imaging, high-throughput screening, 3D imaging, color imaging, and the integration of machine learning. Finally, we explore the application prospects of FPM in digital pathology, drug screening, and other fields and look forward to its future development.

## 2. Technology

### 2.1. Computational Imaging

Against a backdrop that includes the introduction of photosensitive materials in the 18th century, the widespread adoption of actual film photography in the 19th century, followed by the rise of digital backs in the late 20th century and, most recently, the popularity of image processing, computational imaging has swiftly come to the forefront as a widely discussed research area. FPM and algorithms such as SIM and the transport of intensity equation (TIE) belong to the realm of computational imaging. The major advantage of computational imaging algorithms over traditional direct imaging algorithms based on digital or physical principles is their ability to collect information that cannot be directly used for imaging, resulting in improved imaging outcomes [38,39,40]. Moreover, they offer the possibility of systematically optimizing imaging systems. However, there is a trade-off among the FOV, image capture speed, and resolution, creating a triangular dilemma where compromises must be made. In current algorithms like SIM, for instance, image capture speed is high, but the improvement in resolution is relatively modest.

Nevertheless, the progress in a single method has its limitations, and the most promising direction for computational imaging algorithms is the fusion of different algorithmic approaches. In recent years, Li et al. successfully combined TIE with FPM, achieving interference-free synthetic aperture 3D unlabeled microscopy imaging [41]. Xue et al. combined SIM and FPM to achieve rapid high-resolution fluorescence imaging [42]. Computational imaging, as an emerging field, holds extensive prospects for application and can undergo further expansion and discussion.

However, each system has its own strengths and weaknesses. In this section, we focus on discussing the challenges faced by FPM.

### 2.2. Principles of FPM

Fourier ptychographic microscopy represents an advanced imaging technique that overcomes the limitations of traditional microscopy. In traditional microscopy, the maximum resolution is constrained by diffraction and determined by the Abbe diffraction limit formula:(1)$$d_{x,y}=\frac{\lambda }{2NA_{\mathrm{obj}}},$$
(2)$$NA_{obj}=\eta \times \sin\theta .$$
where, $NA_{\mathrm{obj}}$ denotes the numerical aperture of the objective lens, an intrinsic property of the lens itself. The term *d* represents the resolvable feature size, λ is the wavelength of light, η is the refractive index, and θ is the half-angle subtended by the optical objective lens. FPM breaks this constraint by amalgamating principles of computational imaging and Fourier optics. The core concept involves iterative processes in both Fourier space and the spatial domain, establishing a synergy between these two domains. FPM captures low-resolution images at various illumination angles and employs iterative algorithms to reconstruct high-resolution images. It rewrites the original Abbe diffraction limit formula as follows:(3)$$d_{x,y}=\frac{\lambda }{NA_{\mathrm{obj}}+NA_{\mathrm{ill}}},$$
(4)$$NA_{\mathrm{ill}}=\eta \frac{\sqrt{{\left(\frac{m-1}{2}\right)}^{2}d+{\left(\frac{n-1}{2}\right)}^{2}d}}{\sqrt{{\left(\frac{m-1}{2}\right)}^{2}d+{\left(\frac{n-1}{2}\right)}^{2}d}+h^{2}},$$
where η is the refractive index, $NA_{\mathrm{ill}}$ is referred to as the illumination aperture, *d* is the distance from the LED array to the sample, and *m* and *n* represent the numbers of LEDs in both directions. Simultaneously, the core objective of FPM technology is equivalent to finding an optimal object function that best explains the intensity distribution observed by the camera under different angle illuminations. This can be achieved through the following optimization problem:(5)$$\arg\min O(k_{x},k_{y})=\sum m,n\sum_{x,y}|\sqrt{I_{m,n}\left(x,y\right)}-|F^{-1}O(k_{x}-k_{x,m,n},k_{y}-k_{y,m,n})P\left(k_{x},k_{y}\right){\left|\right|}^{2}.$$

The traditional FPM uses the Gerchberg–Saxton algorithm [43], also known as the alternating projection method, to solve the above non-convex optimization problem. The algorithm includes the following steps: initial guess, frequency domain support constraint, spatial amplitude constraint, update object function spectrum, sub-aperture update, and iterative repetition.

The conventional FPM device structure is illustrated in Figure 2g,h, consisting of an array of LEDs and a microscopy system. Data acquisition is accomplished by sequentially illuminating the LEDs and capturing images. The original FPM reconstruction algorithm by Zheng et al. in 2013 has undergone significant developments in the past decade. In 2014, Ou et al. introduced the EPRY-FPM algorithm, capable of simultaneously reconstructing the object function and the pupil function [44]. Zuo et al. proposed an adaptive variable step-length iterative method in 2016, showing enhanced robustness [45]. Subsequent optimization methods based on first-order and second-order gradients were introduced, including the Gauss–Newton method [46], global quasi-Newton method [47], and Wirtinger flow method [48]. In 2022, Wang et al. presented the ADMM-FPM algorithm, which decomposes the FPM reconstruction problem and introduces an adaptive step-size strategy [49]. Comparative analysis in simulations and experiments demonstrated that ADMM-FPM outperforms existing algorithms in terms of robustness and reconstruction quality under noise interference. This algorithm provides a novel and effective reconstruction method for FPM.

The key innovation of FPM lies in its ability to simultaneously achieve a large FOV and high resolution, breaking the conventional trade-off between these factors. FPM leverages the diversity of information obtained from various angles to generate a spectrum of spatial information, ultimately enabling super-resolution imaging. Through iterative optimization of the object function to accurately depict the intensity distribution observed under different angle illuminations, FPM attains impressive imaging capabilities. However, FPM faces challenges such as imaging speed, 3D imaging complexities, and limitations in color imaging. Addressing these challenges and enhancing imaging performance by integrating cutting-edge deep learning with FPM is a topic of active discussion.

Moreover, it is essential to emphasize that FPM holds significant potential in the realms of digital pathology, intraoperative pathology, and biomedical applications. The subsequent section delves into specific research areas related to FPM, focusing on strategies to improve imaging speed, overcome challenges in 3D imaging, achieve full-color acquisition, and harness the synergies between deep learning and FPM. These advancements underscore the transformative impact FPM could have in various cutting-edge technologies and applications, making it a compelling area of exploration in the field of microscopy.

### 2.3. Fast FPM

If the FP process is divided into two parts, image acquisition and image reconstruction, then the overall time of FP can be reduced by reducing the image acquisition time and improving the speed of the reconstruction algorithm. At present, the subsequent iterative reconstruction process is completed in the computer, and it is difficult to quantify its speed. Therefore, the current research on improving FPM speed is focused on reducing the sampling time. Since FP requires a certain overlap of spectra for the collected images, it will be simplified to the traditional phase retrieval process. Obviously, if the image overlap rate is high, more images need to be collected, which will require more sampling time; at the same time, if the overlap rate is too low, the sampling time can be greatly reduced, but the image recovery quality will inevitably be poor. Research on how to minimize FP sampling requirements has continued to develop over the years.

In 2014, Zheng et al. proposed sparsely sampled FP and discussed the minimum overlap rate in the article. The team simulated situations with different overlap ratios, in which the incident wavelength was 632 nm, the pixel size was 2.75 μm, and the objective lens NA was 0.08. The simulation used a 15 × 15 LED array to illuminate the sample at different angles. By adjusting the distance between the LED array and the sample, different spectral overlap ratios are achieved [50]. Shown in Figure 2a1,a2 are the high-resolution intensity and phase images of the simulation input, while (b1)–(d1) are the intensity images reconstructed by FP under different overlap rates, and (b2)–(d2) show the phase diagram under different overlap ratios. In order to quantify the image quality of FP reconstruction under different overlap ratios, root mean square (RMS) errors (i.e., the difference between the ground truth and the recovered image) are defined to quantify them (see Figure 2e). After multiple simulations, the drawn line graph shows that there is not a simple linear relationship between the overlap rate and image recovery quality. At the same time, this image illustrates that successful FP reconstruction requires at least 35% overlap. In the sparse sampling that FP subsequently proposed, one original pixel is divided into four sub-pixels for down-sampling. Only one of the four sub-pixels is updated in each iteration, and the others remain unchanged. At the same time, a down-sampling mask is used to solve the pixel aliasing problem. By selectively updating pixel values in the spatial domain, this solution is able to get rid of the multiple exposure acquisition process in the original FP platform and greatly shorten the acquisition time. In the experiment for this scheme, USAF was used as the sample. The incident wavelength was 0.63 μm, the pixel size was 4.125 μm, the objective lens NA was 0.1, and a 15 × 15 LED array was used to illuminate the sample at different angles. The experiment showed that this scheme is completely feasible.

In 2015, Tian et al. proposed a source-coded FPM technology for mixed lighting, the principle of which is shown in Figure 2f. In the lighting part, it first captures four DPC images (top, bottom, left, and right half-circles) to cover the bright-field LEDs and then uses random multiplexing with eight LEDs to fill in the dark-field Fourier space region. Secondly, in the reconstruction part, it uses a linearly approximated phase solution based on DPC deconvolution as a close initial guess for spatial frequencies within the 2 NA bandwidth [53]. Through the above solution, the number of collected images can be reduced to 21, and the collection time only takes 0.8 s. In the experiment, 21 images (taking 0.8 s) and 173 images (taking 1 min) were collected from a single frame of adult rat neural stem cells for reconstruction and recovery. The reconstruction results show that this method significantly reduces the number of images required and effectively improves the overall speed of FPM while ensuring a certain resolution.

In 2018, Sun et al.proposed high-speed Fourier ptychographic microscopy based on programmable annular illuminations, using only four low-resolution images corresponding to oblique illumination to achieve high-speed imaging results for HeLa cells, the principle of which is shown in Figure 2g. For FPM, low-frequency phase information is difficult to recover. Only when the LED is accurately located at the edge of the objective lens in the frequency domain can the low-frequency phase information be correctly recovered. In order to ensure accurate phase reconstruction, this method first requires precise adjustment of the position of the LED array. Secondly, by lighting only the LED elements on a ring lighting scheme, only 4–12 bright-field raw images are needed to achieve high-precision phase retrieval, significantly reducing data redundancy. An important reason why traditional FPM takes so long is that dark-field images require a longer exposure time (generally, each dark-field image requires 30 ms exposure) to ensure a certain signal-to-noise ratio [51], while this method only uses bright field images, which reduces exposure time to 10 ms per raw image. Under the above premise, the team only needs to light up 4 LEDs at the fastest, which can reduce the collection time to 0.04 s. Along the same lines, AIFPM can bypass the influence and limitations of the camera pixel size and resolve the two closely spaced features with distance of 655 nm.

In 2022, Zuo et al. also proposed an efficient synthetic aperture for phase-less Fourier ptychographic microscopy with hybrid coherent and incoherent illumination (ESA-FPM). Figure 2h shows the hybrid lighting process of ESA-FPM. During the image acquisition process, when collecting bright field images, all LEDs are turned on at the same time, and a single intensity image is collected under incoherent lighting conditions. When collecting dark-field images, sparse sampling is based on central symmetry to ensure improved resolution. The imaging resolution limited by coherent diffraction is achieved with only seven original images. Compared with traditional FPM, the amount of data required is only 1.6%. As lighting methods change, iteration strategies are bound to change as well. During each iteration, the sub-aperture spectrum corresponding to the bright-field illumination is extracted, and its amplitude is updated by the bright-field measurement. The spectrum filtering function is defined as OTF instead of CTF to conform to incoherent imaging. Without sub-aperture scanning, one update fills the spectrum with 2× the coherent diffraction bandwidth of the objective lens. For dark-field illumination, the same dark-field intensity measurement is filtered by CTF to sequentially update two sub-apertures (sub-aperture 1 and sub-aperture 2) of the centrosymmetric distribution [52]. At the same time, in order to avoid falling into a local minimum, an adaptive compensation strategy is added to update the weights.

### 2.4. Full-Color Acquisition

Given the sensitivity of human vision to color information and our innate tendency to classify objects based on their hue, it is common practice to stain biological samples for enhanced recognition and categorization. Therefore, digital pathology places significant emphasis on recovering and acquiring accurate color representation of stained histopathological samples. FPM offers a significant advantage in generating large FOV, high-resolution, and full-color whole-slide images without the need for mechanical scanning. This is achieved through five different methods that ensure full-color image acquisition.

FPM can accommodate both monochrome and chromatic cameras. With a monochrome camera, FPM can retrieve high-resolution images at three different wavelengths (red, green, and blue) and combine them into a single high-resolution full-color image. This process involves sequentially illuminating the slide with red, green, and blue light to obtain the color information, which can be easily achieved using a programmable R/G/B LED array. While there are no additional requirements for overlapping ratios or system environments compared to traditional FPM, it is important to note that the restored full-color images may exhibit coherent artifacts caused by dust particles on the slide or lens. Moreover, color reconstruction can be time-consuming, and intense calibration of the three wavelengths is required. Similar to monochrome cameras, chromatic cameras with Bayer filters do not have additional limitations on overlap ratios or system environments. However, while they offer the advantage of separating three primary channels more efficiently, their larger pixel size often results in lower photographic efficiency. Further, correcting for color leakage remains an important consideration [54].

Significant efforts have been devoted to developing wavelength-multiplexed FPM that utilizes a monochrome camera and simultaneous multi-wavelength illumination [54,55,56,57,58]. While this approach appears promising, reducing acquisition time by approximately two-thirds, it requires an overlap ratio of nearly three times that of traditional FPM [57]. Moreover, the need for sophisticated algorithms to address decoupling issues increases system complexity, making error calibration challenging. Additionally, acquiring three separate low-resolution images in different wavelengths is necessary to prevent converging to similar gray values, further complicating the process [57,58].

It is also possible to apply a deep learning technique to perform unsupervised image-to-image translation of FPM reconstructions, thereby enhancing image quality and color accuracy while decreasing artifacts caused by coherent illumination [59,60]. Specially, a cycle-consistent adversarial network with multiscale structure similarity loss is employed and trained using two sets of unpaired images. Notably, the network’s output closely aligns with the ground truth intensity, and the overall image quality surpasses that of the FPM color image obtained through sequential red, green, and blue illuminations. Also, other than reducing the FPM coherent artifacts, this data-driven approach can shorten the acquisition time of FPM by 67%.

Instead of employing a neural network for image translation, two novel approaches, called color-transfer FPM (CFPM) and color-transfer filtering FPM (CFFPM), have been proposed as an alternative to virtually stain an FPM image [61,62]. Compared with the previous CFPM, CFFPM replaces the original histogram matching process with a combination of block processing and trilateral spatial filtering and both solves the double-coloring problem and improves both the precision and speed of color transfer. CFFPM can perform accurate and fast color transfer for various specimens and for some cases, which can also outperform the sequential conventional methods because of the coherent artifacts introduced by dust particles.

## 3. Applications

### 3.1. Digital Pathology

Digital pathology, regarded as a bridge between basic medicine and clinical diagnosis [63], has absorbed the latest achievements in whole-slide imaging systems for their distinct and efficient observation of the changes in function, metabolism, and morphological structure of organisms. In this regard, FPM demonstrates many advantages and a promising future for digital pathology as a new generation of high-efficiency and advanced imaging modes.

Conventional microscopes have been suffering from the trade-off between FOV and spatial resolution, which in contrast can only achieve high resolution images through high-NA objectives and then stitch them together in the spatial domain to acquire a large FOV image with high resolution, inevitably leading to image artifacts and low efficiency, as well as a limited depth of field along with the high-NA objective [35]. Compared to conventional imaging modes, FPM can simultaneously acquire images of histology sections with both a large FOV and high resolution without any manual post stitches.

Meanwhile, due to the small depth of field of high-NA objectives, constant refocusing is required when the FOV changes to keep the sample in focus, which is time-consuming and cumbersome. FPM allows for digital refocusing during the reconstruction process to address this problem, bringing every small segment of the image into focus [64] and recovering the imaging system’s pupil function to further improve the quality of images [44,65]. As shown in Figure 3a–d, unlike other imaging systems, the digital refocusing capability of FPM allows for imaging of cells at different focal planes to be achieved on the same plane, despite surface unevenness, thereby facilitating the identification and enumeration of circulating tumor cells (CTCs). Additionally, the large FOV provided by FPM avoids the issues of CTCs failing to be counted or a single event being counted more than once during the stitching process [66]. Similarly, a large FOV of cell analysis can also be performed on the FP-recovered high-content images. As demonstrated by the FPM results of thyroid fine needle aspiration cytology samples in Figure 3e, it is evident that all non-overlapping cells in the diagnostic cluster can be seen in sharp focus, regardless of the thickness of the samples [67].

Another important application of FPM in digital pathology is to extract the scattering characteristics of tissue samples from the recovered phase information. This scattering characteristic is directly related to the refractive index (RI) of the tissue and helps distinguish healthy cells from cancerous cells in digital pathology diagnoses [68]. As shown in the zoomed-in views of Figure 3f, the recovered phase can be used to obtain the local scattering and reduce the scattering coefficients of the specimen. Also, FPM’s captured quantitative images with high contrast are an ideal method for non-invasively monitoring biological samples and analysis. As shown in Figure 3g, it is evident that the high-contrast areas corresponding to the nuclei in the hematoxylin and eosin-stained (H&E) renal tissue slide are distinguishable as well in the stain-free FPM image [69]. At the same time, the unstained kidney tissue section images provided by FPM can also obtain quantitative phase information while avoiding the influence of staining color and intensity on imaging effects that comes along with the markers. While taking cell segmentation and the counting of several specific types into consideration, FPM can provide higher overall image quality compared to fluorescent imaging and DPC, with higher contrast and clear, visible, fine details [70].

### 3.2. Drug Screening

As shown in Figure 4a, drug screening has gone through five different stages of development: physiology-based high-throughput screening (HTS), chemistry-based HTS, high-content screening (HCS), high-screening imaging (HSI), and high-throughput HCI (HT-HCI). HTS is a well-established process for lead discovery in pharma and biotech companies [71], which is shown in Figure 4b, and it comprises the screening of large chemical libraries for activity against biological targets via the use of miniaturized assays and large-scale data analysis [72] such as flow cytometry, which enables the simultaneous quantitative analysis in individual cells [73]. However, as shown in Figure 4c, this conventional method suffers from time, cost, and quality, and it has been difficult to adapt it to the research of related therapeutic drugs for polygenic diseases and viral infections [72]. Compared to HTS, HCS combines the efficiency of high-throughput techniques with the ability of cellular imaging to collect quantitative data from complex biological systems [74,75,76]. Among them, FPM-based screening systems enable simultaneous imaging and analysis of the morphology of cells, greatly enhancing the experimental scale and efficiency of drug screening.

Currently, cell-based high-throughput screening systems based on commercial porous plates can only provide limited and rough descriptions of the samples’ properties [77,78,79,80]. Given the high-throughput advantage of FPM, which can provide rich information about cell cultures such as morphology, integrity, and viability, a 6-well and 96-well cell culture imaging and drug screening system based on parallel FPMs have been sequentially developed and reported [81,82,83], called 6 Eyes and 96 Eyes, respectively. Figure 5 demonstrates a parallel microscopy system that can simultaneously image all wells on a 96-well plate via FP and can perform dual-channel fluorescence imaging at the native resolution of the 96 objectives.

Although excellent methods exist to image quantitatively complex processes such as protein–protein interactions by fluorescence resonance energy transfer [84,85] or the dynamics of the turnover of fluorescent proteins on cellular organelles [86,87,88], their application for object recognition algorithms for ‘real time’ image analysis in high-throughput microscopy still needs to be developed [89]. The combination of increasingly advanced artificial intelligence and high-throughput FPM not only successfully meets this requirement but also promotes the development of drug screening. The number of white blood cells (WBCs) is a valuable indicator for diagnosing or predicting various diseases [90,91,92,93,94,95]. An automatic counting algorithm was developed to accurately distinguish WBCs from images recovered from FPM with 95% accuracy [96]. Moreover, machine learning (random forest) and deep learning (VGG16) models can be used to diagnose the infection status of thousands of red blood cells within a single FOV, achieving 91% and 98% specificity, respectively [97].

### 3.3. Label Free

In vitro microscopy is crucial for studying physiological phenomena in cells. For many applications, such as drug discovery [76], cancer cell biology [98], and stem cell research [99], the goal is to identify and isolate events of interest. Free from the adverse effects of staining reagents on cell viability and signaling in existing imaging techniques [100,101,102,103,104], high-throughput FPM has been demonstrated for long-term label-free observation and quantitative analysis of large cell populations without compromising the spatial and temporal resolution, and it has shown great promises in important applications in drug discovery, personalized genomics, cancer diagnosis, and drug development [35,105,106].

In contrast to fixed slides, live samples are continuously evolving at various spatial and temporal scales. Faster capture times would not only improve the imaging speed, but also allow studies of live samples, where motion artifacts degrade results. A new source coding scheme was proposed and achieved 0.8 NA resolution across a 4× FOV with sub-second capture times [53]. As shown in Figure 6a, a HeLa cell is undergoing mitosis and dividing into four cells, and the whole process stays in focus across the FOV where sub-cellular features are visible and the dynamics of actin filament formation can be tracked over time. Additionally, both fast-scale dynamics and slow-scale evolution of adult rat neural stem cells with details at both the sub-cellular level and across the entire cell population can be seen in Figure 6b. Moreover, in order to address the time-varying aberration and focus drifts in long-term live-cell imaging, a computational quantitative phase imaging (QPI) method based on annular illumination FPM was reported, in which the annular matched the illumination configuration. As shown in Figure 6c, the mother cell underwent three mitoses across 27 h and was eventually divided into four individual daughter cells with high imaging quality [107], which reveals the ability of the FPM-based approach to correct temporally varied aberrations and secure the imaging performance for a long-term longitudinal study [108,109,110,111,112].

### 3.4. Three-Dimensional Imaging

In the biomedical areas, high-resolution 3D imaging attracts wide attention due to more information contained in the cells or tissues [113,114,115]. However, it is extremely difficult for the traditional microscope to reconstruct thick samples in 3D. However, for FPM, due to the long depth of field (DOF) [35] and other characteristics, there are three main methods to realize 3D imaging: digital refocusing, multi-slice, and optical coherence tomography.

It is necessary to perform mechanical scanning to gain the 3D volume imaging sample for the traditional microscope. However, there are defocused artifacts due to the interference of the upper and lower planes outside the DOF, and the DOF is defined by:(6)$$DOF=\frac{\lambda }{NA^{2}}+\frac{\lambda }{M\times NA}e,$$
where *e* is the minimum resolvable distance of the detector and *M* stands for magnification [116]. Luckily, FPM can achieve digital refocusing, leveraging the feature of coherent light. FPM can “digitally refocus” images by numerically zeroing out the defocus aberration. Digital refocusing refers to the algorithm focusing method according to the characteristics of the data after the completion of imaging data acquisition. Zheng and Bian et al. multiply the phase term, which is related to the defocus distance of the pupil function. Then, this pupil function is as follows:(7)$$P^{\prime }\left(k_{x},k_{y}\right)=P\left(k_{x},k_{y}\right)e^{jz\sqrt{{k_{0}}^{2}-{k_{x}}^{2}-{k_{y}}^{2}}},{k_{x}}^{2}+{k_{y}}^{2}<{\left(k_{0}NA\right)}^{2},$$
where *z* denotes the defocus amount.

This is performed by determining and removing the defocus aberration during the iterative phase retrieval process to extend the DOF. Using a 2× apochromatic objective lens (Plan APO, 0.08 NA, Olympus), the FPM can digitally refocus to extend the depth of field to 300 μm [35]. This operation allows us to eliminate the defocus aberration in each pupil function at each illumination angle. However, this algorithm requires knowledge of the out-of-focus distance *z* as a priori information. Otherwise, it often consumes a lot of computational time to search for a suitable out-of-focus distance. Furthermore, Zhou et al. demonstrated that the process of refocusing in FPM cannot be separated from the iterative phase retrieval procedure without prior information [117]. Zhang et al. proposed a fast digital refocusing and DOF-extended FPM strategy by taking advantage of image lateral shift caused by sample defocusing and varied-angle illuminations [118]. The degree of lateral shift is directly related to the amount of defocusing and the tangent of the illumination angle. Instead of using a time-consuming optimization search to find the best defocus distance, this method allows for precise and quick determination of the defocus distance for each subsection of the sample by calculating the relative lateral shifts corresponding to different oblique illuminations. The effectiveness of this approach in achieving rapid digital refocusing and extending the DOF was confirmed through practical experiments using a USAF chart (Figure 7a). Zhang et al. have employed coded-detection methods to separate the sample-focusing step from the iterative phase retrieval process. By obtaining the retrieved object exit wavefront, it becomes possible to propagate it to various axial positions and employ a focus metric to determine the most suitable focal position. For stained samples, focus metrics based on intensity can be applied, whereas for unstained samples, focus metrics based on phase are more appropriate [119,120,121]. However, the maximum length of the depth-of-field extension achieved by digital refocusing does not exceed the coherence length. The system satisfies the approximation of a 2D thin object only if the thickness of the sample satisfies certain conditions [122]. Similar to stacked diffraction imaging, FPM requires that the thickness of the observed sample generally does not exceed 10 μm, which severely limits the application of FPM in 3D microscopy imaging [123]. Zhang et al. introduced a new imaging technique called ptychographic structured modulation for super-resolution microscopy [124]. They used a thin diffuser to manipulate light from the sample, allowing high-resolution details to be captured. Ptychographic structured modulation overcomes the diffraction limit, handles thin samples better, and provides accurate complex object contrast. The digital propagation of the recovered complex wavefront to two different layers. Additionally, they employed an aperture-scanning FP setup to retrieve the complex hologram of extended objects [125]. This reconstructed hologram is digitally propagated through various planes along the optical axis to investigate the 3D structure of the object in Figure 7c.

In view of the limitation of sample thickness, many studies follow the example of real-space ptychography imaging and combine a multi-slice model with FPM technology [34,126,127,128,129]. Tian et al. introduced the concept of multi-slice modeling in FPM [126]. This innovative method enables the reconstruction of 3D sample information at different depths along the optical axis, effectively surpassing the limitations of traditional microscopes. In this multi-slice approach, the sample is divided into thin slices, and wave fields pass through each of them. However, as the number of slices increases, the computational workload grows significantly, impacting computational efficiency. Moreover, achieving high reconstruction accuracy, particularly when using high-NA objectives, can be challenging due to light scattering in various directions, including sideways and backward. Chowdhury et al. implement a new 3D RI microscopy technique that utilizes a computational multi-slice beam propagation method to invert the optical scattering process and reconstruct high-resolution (NA > 1.0) 3D RI distributions of multiple-scattering samples. The method acquires intensity-only measurements from different illumination angles and then solves a nonlinear optimization problem to recover the sample’s 3D RI distribution. This is also demonstrated by the reconstruction of samples with varying amounts of multiple-scattering. A 3D RI of a 3T3 fibroblast cell is shown in Figure 7b.

Unfortunately, refocusing does not remove light from areas above and below the plane of interest. However, the multi-slice method does not directly account for back-scattered light. Its projection approximation also assumes that the lateral divergence of the optical field gradient at each slice is zero. This longstanding problem has inspired a number of solutions. Horstmeyer et al. performed diffraction tomography (DT) using standard intensity images captured under variable LED illumination from an array source, termed Fourier ptychographic tomography (FPT) [130]. Diffraction tomography, initially proposed by Emil Wolf [131], combines the theory of X-ray tomography and holography to estimate the 3D RI distribution of an object under weak scattering (single scattering) approximation. Horstmeyer et al. employed diffraction tomography, utilizing intensity measurements acquired with a standard microscope and an LED illuminator. This system achieves a lateral resolution of about 400 nm, operating at the Nyquist–Shannon sampling limit. Regarding axial resolution, it reaches 3.7 μm at the same sampling limit. Their experiments extended up with a maximum axial depth of 110 μm along the z-axis. Remarkably, they successfully demonstrated quantitative measurements of the complex RI across various thick specimens featuring contiguous structures. Michael Chen, Laura Waller, et al. introduced a precise and computationally efficient 3D scattering model known as the multi-layer Born model. They harnessed this model to recover the 3D RI of thick biological specimens [132]. To implement MLB effectively, they integrated it into a phase tomography framework, relying solely on intensity-only images acquired using the FPM. In a separate study, Sandro et al. applied FPT to coherent anti-Stokes Raman scattering imaging. Their work demonstrated that complex third-order susceptibilities can be reconstructed in 3D using synthetic data and a limited number of wide-field coherent anti-Stokes Raman scattering images. Zuo et al. developed Fourier ptychographic diffraction tomography (FPDT), an approach for generating high-resolution 3D RI images across substantial volumes. FPDT accomplishes this by utilizing low-NA intensity measurements and incorporating high-angle dark-field illumination [133]. The results are truly impressive, featuring a wide FOV measuring 10× FOV of 1.77 mm^2^, with exceptional 390 nm lateral resolution, 899 nm axial resolution, and a depth of focus of approximately 20 μm, as shown in Figure 8a. Li et al. have introduced an innovative label-free 3D microscopy technique called transport of intensity diffraction tomography with non-interferometric synthetic aperture (TIDT-NSA) [41]. This approach allows for the retrieval of the 3D RI distribution of biological specimens from 3D intensity-only measurements acquired at various illumination angles, enabling incoherent-diffraction-limited quantitative 3D phase-contrast imaging. Utilizing an off-the-shelf bright-field microscope equipped with a programmable LED illumination source, TIDT-NSA achieves an impressive imaging resolution of 206 nm laterally and 520 nm axially when using a high-NA oil-immersion objective. One notable advantage of TIDT-NSA is its ability to eliminate the need for a matched illumination condition (analyticity condition) typically required in 2D Kramers–Kronig relations. Instead, it leverages 3D intensity transport, which provides direct access to the object’s frequency content within the generalized aperture. Furthermore, a unified transfer function theory of 3D image formation has been derived, establishing a link between the 3D object function (scattering potential) and the 3D intensity distribution under first-order Born/Rytov approximations. This enables direct non-interferometric 3D synthetic aperture imaging in the Fourier domain, as shown in Figure 8b. Building upon Li’s research, Zhou et al. have introduced transport-of-intensity Fourier ptychographic diffraction tomography(TI-FPDT) as a solution to address challenging issues in microscopy [134]. TI-FPDT combines ptychographic angular diversity with additional “transport of intensity” measurements. This approach takes advantage of the defocused phase contrast to circumvent the stringent requirement on illumination NA imposed by the matched illumination condition. Consequently, TI-FPDT effectively mitigates issues related to reconstruction quality deterioration and RI underestimation seen in conventional FPDT. These improvements are evident in high-resolution tomographic imaging, such as the imaging of USAF targets, as shown in Figure 8c. Habib et al. present a parallel implementation of a synthetic aperture in TIDT (PSA-TIDT) using annular illumination [135]. This matched annular illumination setup yields a mirror-symmetric 3D optical transfer function, signifying the analyticity in the upper half-plane of the complex phase function. This unique feature enables the recovery of the 3D RI from a single-intensity stack. To validate the effectiveness of PSA-TIDT, they conducted rigorous experimental tests. These experiments included high-resolution tomographic imaging, resulting in the reconstruction of the 3D RI rendering of Henrietta Lacks (HeLa) cells.

### 3.5. Deep Learning

In recent years, applying deep learning methods to the reconstruction of FPM has become a common trend, as these methods have been demonstrated to be powerful tools for solving data problems, including the phase retrieval in FPM reconstruction [37] and in all QPI technologies [136]. Three possible research directions, divided by their roles in the FPM to improve the performance of the technology, have been explored. The first group worked on the research point that aims to infer high-resolution images from the easily obtained low-resolution intensity or phase images, entirely considering the FPM reconstruction as a data proposal problem [137,138,139,140,141,142]. Actually, data-driven image reconstruction techniques based on deep learning have gained tremendous success in solving complex inverse problems. Figure 9a shows the reconstruction results of a deep learning approach over an imaging course of 4 h and explicitly shows the significant morphological changes during the process. The second group sought the incorporation of the physical model and the deep learning framework, which made the whole model more interpretable [143,144,145,146,147]. Such physical models adopt a modular design and feature fusion to complete the image reconstruction. Compared with the classical method, this method greatly saves running time and speed and has good performance in the evaluation index, which provides a new improvement idea for future FPM reconstruction. The third group tried to model the forward imaging using a neural network and perform optimization via a network training process, which utilizes a forward pass to model the real imaging process of the actual FPM reconstruction process [60,148,149,150,151]. Figure 9b compares the reconstruction results of intensity and phase images between a forward imaging neural network and the traditional algorithms and other neural networks. Figure 10 shows the reconstruction results of another forward imaging neural network called FINN-P with its intensity images, phase images, and pupil recovery.

All these groups have obtained excellent progress recently. For the first group, different neural networks have been proposed. Researchers utilized DeUNet to generate the Cross-level Channel Attention Network, which used the coding method for denoising of measurements [152]. A residual transfer network was used to overcome the gradient explosion [49], making the feature information more complete and efficient. The incremental up-sampling reconstruction network has higher image quality, lower computational complexity, and a shorter running time. As a combination with the neural network and the previous phase retrieval algorithms, LWGNet tried to combine the Wirtinger gradients with a neural network and enhanced the gradient images by convolution modules, which may significantly bring down the cost of FPM imaging setup [153]. A double-flow convolution neural network that separates the network data flow into two branches has successfully reduced the processing time from 167.5 to 0.1125 s with fine generalizability and little dependence on the morphological features of samples [154].

**Figure 10 cells-13-00324-f010:**
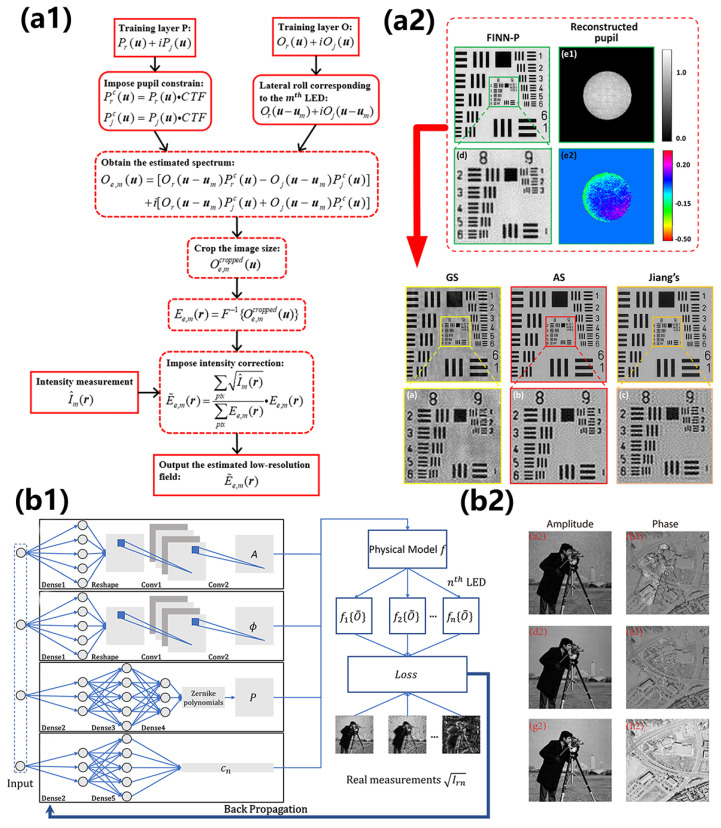
Frameworks and reconstruction of deep learning methods working on FPM. (**a1**) Overall workflow of FINN-P method. (**a2**) Reconstruction results in an enlargement of FINN-P and the amplitude and phase of the pupil function recovered through FINN-P. Reproduced from [150], CC BY 3.0. (**b1**,**b2**) Structure and reconstruction results of deep learning framework based on deep image priors. Reproduced from [155], CC BY 3.0.

For the second group, new physical models like SwinIR have been introduced into FPM for completing the reconstruction of images by introducing modular design and feature fusion [49]. Through a series of simulation experiments on ideal images and real images, it is proved that this physical model has better reconstruction quality than the comparison algorithm. A similar idea of feature fusion was also demonstrated in the Residual Hybrid Attention Network [156], which successfully simulated network noises. Another physics-inspired model with a deep learning framework was also proven efficient with a low overlap condition as it reached a super-resolution factor of γ = 5 with only 37 images [157]. A neural network with physics-based channel attention was also used for FPM reconstruction to enable the adaptive correction of the spatial distribution of LED intensity [158]. For the third group, setting the real part and imaginary part of the object, as well as the different and irregular positional deviations of each aperture as the weights of the convolutional layer, has been proven to be effective in accurately finding the aperture position and improving the reconstruction quality [159]. Aside from these previous research points, a deep image prior was also proposed in the FPM. As a deep learning method, it is able to provide a starting point, a prior, which allows a great reduction in the artifacts, especially for an illumination array of limited size [160]. Unlike the traditional training type of deep neural network that requires a large labeled dataset, an FPM with untrained deep neural network priors does not require training and instead outputs the high-resolution image by optimizing the parameters of neural networks to fit the experimentally measured low-resolution images [155]. With the application of a priori information, deep neural networks can evidently reduce the requirement for practical applications of FPM [161].

## 4. Outlook

Although FPM has demonstrated its versatility as an imaging technique, there are still imperfections in FPM to date that affect new scientific discoveries. Here, we explore the exciting possibilities and future directions for FPM in technological domains.

Algorithms: New algorithms for FPM have continuously emerged in recent years with better speed of convergence and robustness. Wu et al. proposed an aberration correction method; by applying an adaptive modulation factor in the FPM reconstruction framework, this algorithm performed better in terms of robustness and convergence for eliminating hybrid aberrations [162]. In the quest for higher resolution, FPM algorithms are increasingly employing advanced reconstruction techniques, such as sophisticated phase retrieval algorithms and non-linear optimization approaches. Deep learning methods are also being integrated into FPM algorithms to enhance noise reduction and adapt to challenging imaging conditions.Speed and efficiency: Pioneering swifter and more efficient FPM algorithms alongside optimized hardware configurations could render the technique significantly more viable across a spectrum of applications, including real-time imaging. Notably, the development of real-time imaging systems that incorporate deep learning can revolutionize FPM applications. These systems can rapidly analyze acquired data and make real-time decisions based on FPM-generated information, opening up new possibilities in dynamic and time-sensitive scenarios.Polarization: The properties of polarization add another dimension to FPM. Polarization-sensitive FPM can provide enhanced contrast and resolution, particularly in biological and materials science applications, where the orientation and anisotropic properties of samples are of critical importance. Although instances can be found within the broader field of ptychography, there has not been a documented instance of polarization imaging utilizing FPM, although instances can be found within the broader field of ptychography.3D imaging: FPM is embarking on a promising journey towards accurate 3D reconstructions of samples without the need for scanning. This approach opens up new horizons for researchers, enabling them to explore complex structures in a non-destructive and time-efficient manner. Although numerous endeavors have already expanded the frontiers of FPM to reconstruct 3D images from thick samples, as previously mentioned, further endeavors are essential to comprehensively grasp and model the intricacies of multiple scattering.Multimodal imaging: The fusion of FPM with other imaging techniques, such as fluorescence microscopy or spectroscopy, offers the potential for multimodal imaging. This approach enables the simultaneous capture of multiple types of information from the same sample, providing a holistic view of the specimen under study. By seamlessly integrating complementary techniques, researchers can gain deeper insights into the composition and behavior of complex samples.

Meanwhile, it is worth noting that FPM offers a unique array of advantages that make it an appealing technique for a wide range of applications. Below, we provide a partial list of these potential applications.

Biomedical applications: FPM can address critical challenges in disease diagnostics, drug development, and cellular analysis by providing comprehensive datasets that facilitate a more nuanced understanding of biological specimens. The growing interest within radiology, pathology, and various medical domains centers on automating image-based diagnostic processes through machine learning techniques. FPM presents numerous advantages compared with conventional imaging methods, including heightened throughput resulting from an enhanced synthetic aperture, coupled with its phase sensitivity, which has the potential to elevate the accuracy of automated diagnostic decisions.Extension in EUV regimes: While FPM has made significant strides in visible and near-infrared wavelengths, its extension into extreme ultraviolet regimes offers unprecedented opportunities. The marriage of FPM with EUV sources holds great potential for applications in semiconductor lithography manufacturing by providing nanoscale imaging capabilities critical for quality control and defect detection. FPM can be seamlessly integrated with other EUV spectroscopy techniques, enabling simultaneous imaging and spectral analysis. This approach is invaluable for studying nanoscale materials’ electronic properties, characterization, and chemical composition.Miniaturization and accessibility: To unlock the full potential of FPM, it must become more accessible. Future directions include making FPM systems compact, affordable, and readily available for diverse applications and settings. Designing compact and portable FPM systems that can be deployed in resource-limited settings is a critical step toward democratizing this technology. Such systems have the potential to empower researchers, doctors, and technicians in remote locations.Automated analysis: Machine learning techniques are increasingly finding their way into microscopy. Leveraging machine learning algorithms for automated image analysis and diagnosis represents a transformative direction for FPM. Particularly in fields like pathology and quality control, FPM combined with AI has the potential to streamline processes and improve accuracy.

## 5. Conclusions

This article outlines the development history of FPM over the past 10 years and demonstrates the characteristics and limitations of FPM in detail by comparing it with other imaging technologies. Through some current application prospects, some future development directions are proposed.

Even though FPM technology has been developing for a decade, it still faces three major challenges. Firstly, in the process of image reconstruction, block processing is indispensable due to the influence of the vignetting effect. Compared to traditional mechanical scanning and stitching methods, digital stitching still encounters issues of stitching artifacts and uneven color distribution. This severely limits the clinical application of FPM. Secondly, due to the high computational power requirements, FPM heavily relies on GPU parallel computing. In comparison to CPU-based scanning and stitching full-tile systems, FPM loses the advantages of time efficiency and low cost. Thirdly, constrained by factors such as detector sampling rate, flux, sensitivity, frame rate, etc., current FPM imaging systems mostly use 4× objectives, which cannot meet the requirements of a standard sliced 25 × 75 mm full-tile field of view. While using higher magnification objectives for scanning, it reintroduces the problems of traditional scanning and stitching.

Despite these challenges, FPM holds immense promise for the future of imaging technology. It effectively translates the formidable hardware demands of high-resolution microscopy imaging into challenges that can be conquered through computational means. Advancements in resolution, large FOV, phase recovery, integration with other imaging modalities, and the incorporation of machine learning techniques are poised to revolutionize various domains. By addressing challenges and fostering accessibility, FPM is on the cusp of becoming an indispensable tool in scientific research and industrial applications. The horizon for FPM gleams with potential, and its journey of transformation has only just begun.

## Figures and Tables

**Figure 1 cells-13-00324-f001:**
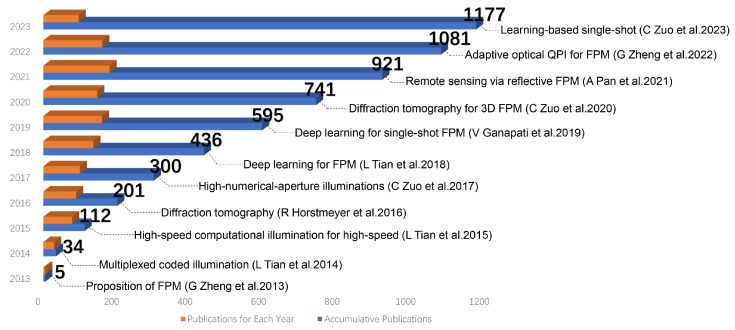
The advancement of FPM has seen remarkable growth in the number of related publications since its inception in 2013. Notable milestones in its development have been highlighted.

**Figure 2 cells-13-00324-f002:**
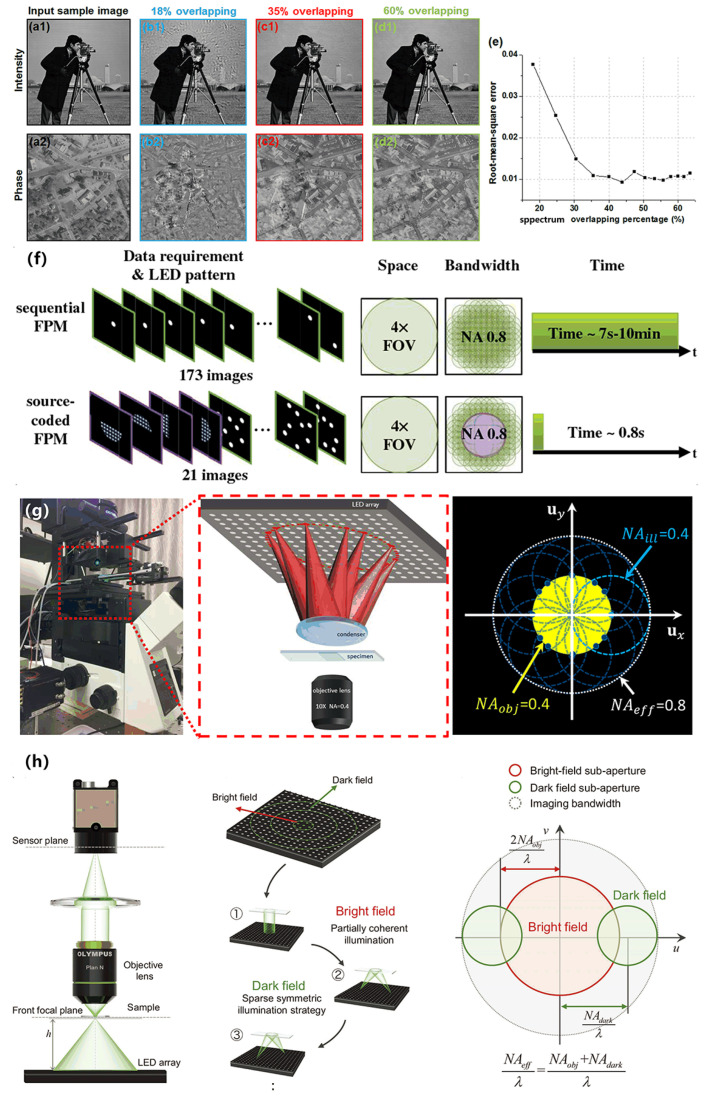
(**a1,a2**) Input high-resolution intensity and phase profiles of the simulated complex sample. Reproduced from [50], CC BY 3.0. (**b1**–**d2**) FP reconstructions with different spectrum overlapping percentages in the Fourier domain. (**e**) RMS errors of the FP reconstructions versus the spectrum overlapping percentages. Reproduced from [50], CC BY 3.0. (**f**) Comparison between sequential FPM and source-coded FPM. Reproduced from [50], CC BY 3.0. (**g**) The experimental setup involves an LED array board, a cemented doublet condenser, an Olympus IX73 microscope with an Olympus UPlanSApo 10× (0.40 NA) objective lens, and a scientific CMOS camera. Reproduced from [51], CC BY 3.0. (**h**) Hybrid illumination modes of ESA-FPM. Reproduced from [52], CC BY 3.0.

**Figure 3 cells-13-00324-f003:**
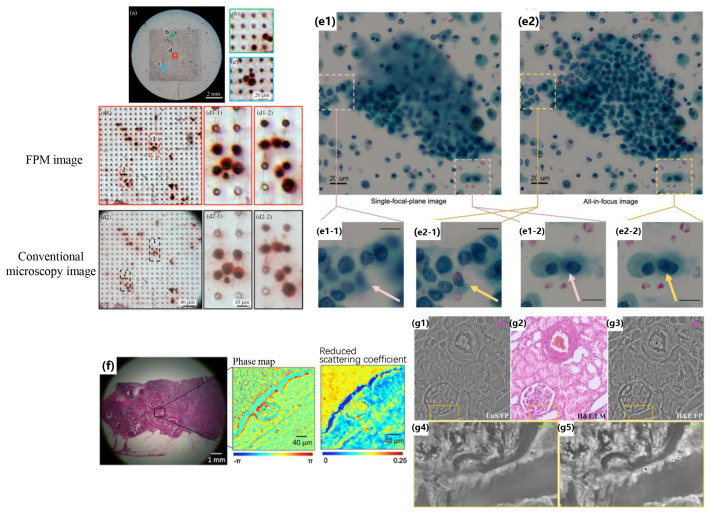
(**a–d2-2**) Full FOV color image of the entire microfilter containing captured tumor cells. Magnified regions indicate the level of detail of cell morphology that can be seen. Reproduced from [66], CC BY 3.0. (**e1**–**e2-2**) Comparison between a single-plane-of-focus image with an all-in-focus FPM image. Single-focal-plane color image reconstructed from FPM. Note the large areas of out-of-focus image in the diagnostic cluster that results from the thickness of the preparation. All-in-focus color image reconstructed from FPM. Reproduced from [67], CC BY 3.0. (**f**) The phase map shows the optical path length delays introduced by the specimen, and the reduced scattering coefficient map quantifies how much light has been scattered by the specimen. Reproduced from [68] by permission from Elsevier: Computerized Medical Imaging and Graphics © 2015. (**g1**–**g5**) Unstained and H&E-stained image comparison. Reproduced from [69], CC BY 3.0.

**Figure 4 cells-13-00324-f004:**
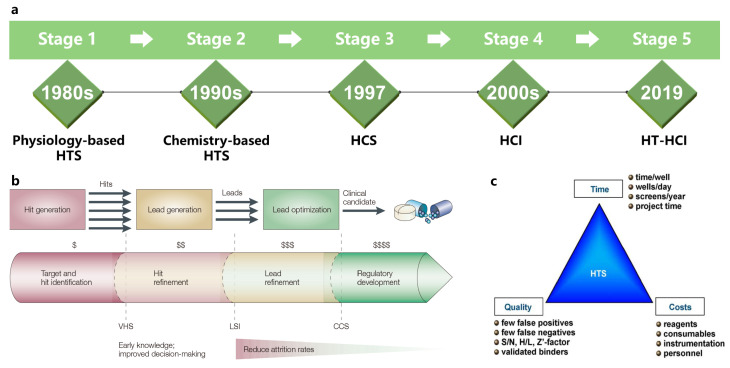
(**a**) Five stages of massive drug screening. (**b**) Progress for lead discovery and corresponding stage-by-stage quality assessment to reduce costly late-stage attrition. Reproduced from [71] by permission from Springer Nature: Nature Reviews Drug Discovery © 2003. (**c**) The optimization process for successful HTS and the key success factors for modern lead discovery via HTS, namely, time, costs, and quality. Reproduced from [72] by permission from Elsevier: Current Opinion in Pharmacology © 2009.

**Figure 5 cells-13-00324-f005:**
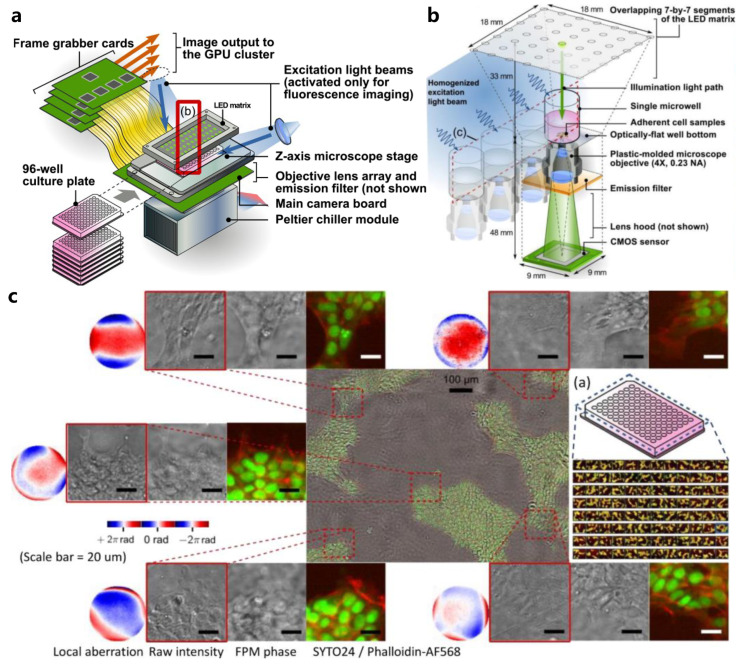
(**a**) General hardware. Individual plates are loaded from the front. Reproduced from [83], CC BY 3.0. (**b**) The imaging module consists of 96 repeating units of compact miniaturized microscopes packed in a 9 mm × 9 mm × 81 mm space, where they all share the same light source. Reproduced from [83], CC BY 3.0. (**c**) Final performance of 96 Eyes with dual-channel fluorescence imaging. Reproduced from [83], CC BY 3.0.

**Figure 6 cells-13-00324-f006:**
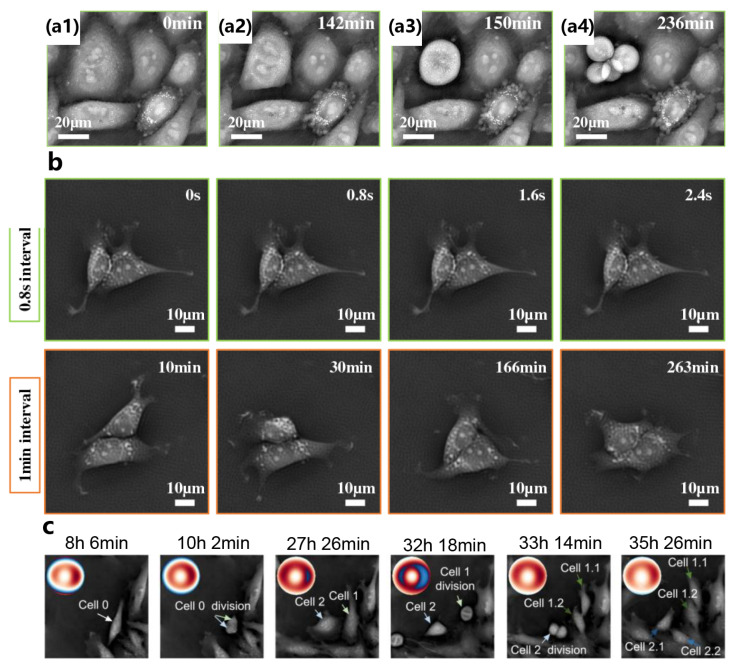
(**a1**–**a4**) A small zoomed-in area of confluent cells in which one cell is dividing into multiple cells. Reproduced from [53], CC BY 3.0. (**b**) A small zoomed-in area. Top: successive frames at the maximum frame rate (1.25 Hz). Bottom: sample frames across a longer time lapse (4.5 h at 1 min intervals). Reproduced from [53], CC BY 3.0. (**c**) The three-generation cell division process from 8 h 6 min to 35 h 6 min. Reproduced from [107], CC BY 3.0.

**Figure 7 cells-13-00324-f007:**
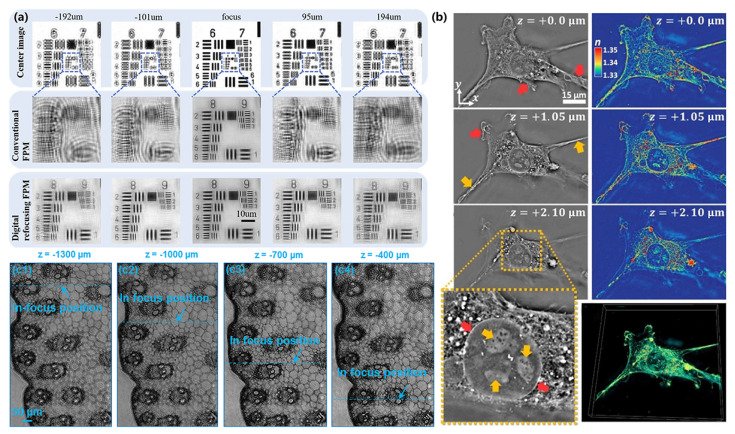
(**a**) High−resolution reconstruction results of conventional FPM and the Zhang et al. proposed method with different defocus distances. Reproduced from [118], CC BY 3.0. (**b**) These images present reconstructed images of fibroblast cells’ RI using the MSBP model and showcase grayscale and colored lateral cross-sections at various depths, highlighting cellular structures and dynamic membrane protrusions. Yellow arrows denote long filopodial extensions, while red arrows indicate broader lamellipodia with membrane ruffling at z = +1.05 μm. They provided a zoom-in of this region at z = +2.10 μm to emphasize the nuclear envelope (red arrows), internal nucleoli (yellow arrows), and the outer region of the optically-dense cellular structure Reproduced from [126], CC BY 3.0. (**c**) 3D holographic refocusing using the aperture-scanning FP. The recovered sections at (**c1**) z = −1300 μm, (**c2**) z = −1000 μm, (**c3**) z = −700 μm, and (**c4**) z = −400 μm. Reproduced from [125], CC BY 3.0.

**Figure 8 cells-13-00324-f008:**
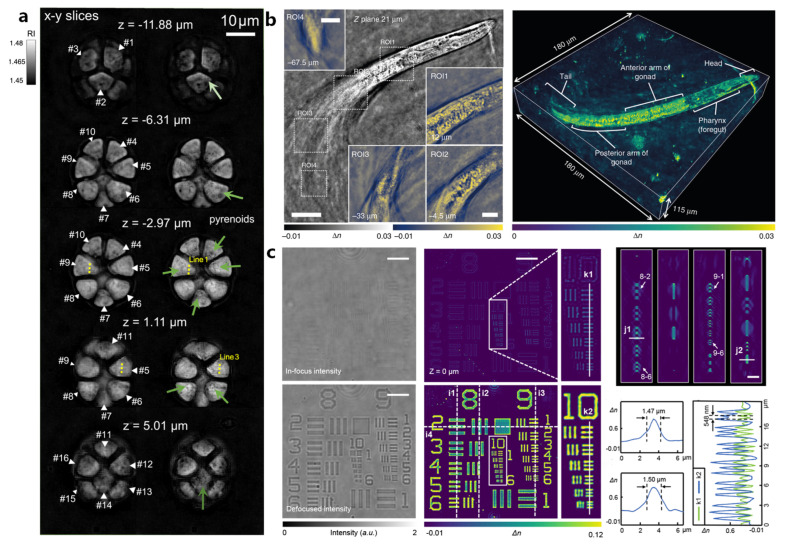
(**a**) The images reveal the inner architecture of algae, illustrating how different cells are interconnected in 3D space. White arrows indicate individual cells, while green arrows highlight subcellular details, such as the periphery of pyrenoids, particularly visible in dark-field images obtained through the FPDT technique. Reproduced from [133], CC BY 3.0. (**b**) Full FOV and different tomogram ROIs at different positions and axial planes to illustrate the recovered RI slice results of *C. elegans* and 3D RI rendering of *C. elegans* worm over a volume of 180 μm × 180 μm × 115 μm. Reproduced from [41], CC BY 3.0. (**c**) In-focus and defocused intensity images of phase resolution target at vertical coherent illumination. Reproduced from [134], CC BY 3.0.

**Figure 9 cells-13-00324-f009:**
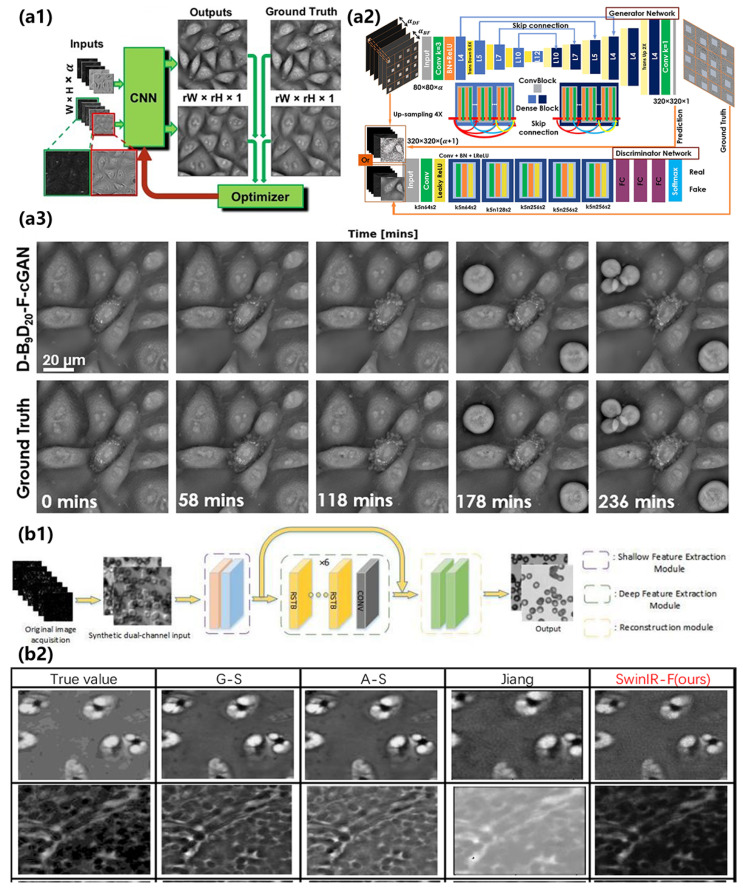
Frameworks and reconstruction of deep learning methods working on FPM. (**a1**) The deep learning framework of inferring HR images from LR images. (**a2**) Details of the CNN structure in (**a1**). (**a3**) Frames of the reconstructed high-SBP phase images observing significant morphological changes over a course of 4 h. Reproduced from [142], CC BY 3.0. (**b1**,**b2**) Structure and reconstruction comparison of deep learning framework based on a physical model called SwinIR. Reproduced from [49], CC BY 3.0.

## Data Availability

Not applicable.

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
