# Peer review of "Fourier Ptychographic Microscopy 10 Years on: A Review"

_cells, 2024, doi:10.3390/cells13040324_

Round 1

Reviewer 1 Report

Comments and Suggestions for Authors

Fourier ptychographic microscopy 10 years on: A review
Fannuo Xu, Zipei Wu, Chao Tan, Yizheng Liao, Zhiping Wang, Keru Chen and An Pan
In this manuscript, the progress in Fourier ptychographic microscopy over the last decade has been reviewed. Conventional microscopes have been suffering from the trade-off between field-of-view (FOV) and spatial resolution, which in contrast can only achieve high resolution images through high-NA objectives and then stitch them together in the spatial domain to acquire a large FOV image with high resolution, inevitably leading to image artifacts and low efficiency, as well as a limited depth of field coming along with the high-NA objective (lines 327-331). I totally agree that this is a very important issue, and Fourier ptychographic microscopy has a great potential to improve imaging process.
This is a solid review discussing the advantages and limitations of FPM in detail by comparing it with other imaging technologies. Biomedical applications, deep learning methods are included. The manuscript is well written, logically organized and properly divided into several sections. Minor corrections :
1. Line 1, Abstract “Fourier stack microscopy (FPM) emerged as…” It seems to be a misprint, change to “Fourier Ptychographic microscopy (FPM) emerged as…”
2. I did not find what is “IIM“(line 82). Probably, it is a good idea to create a list of abbreviations at the beginning or at the end of the manuscript.
3. Eq.1 (page 4), NAobj and NAill are not specified. Probably, it is related to “numerical aperture” but still needs to be identified.
4. Eq.2 (page 4). The equal sign “=” is missing.
5. Fig. 2 (page 5) is difficult to follow. It would be better to split this figure into several figures and put each of them right after the corresponding paragraph describing them.
6. Fig. 2i-2j (lines 266-269) is missing.
7. RI (line 351) HE (line 357) are not specified.
8. Line 367, why high-screening imaging is HCI, and not HSI ?
9. Line 422, QPI is not specified.
10. Line 433, DOF is not specified. Definitely, it is not degree of freedom. Maybe “depth of field” ?
11. Line 438, “…where is the minimum resolvable distance of detector” something is missing.
12. Line 474, “…in 3D microscopy and imaging. Imaging [121]”. A misprint ?
13. Line 501, “…shown in Figure 3.4(b).” change to “shown in Figure 7.4(b).”
14. Line 572. “Figure 10(a) shows…” change to “Figure 9(a) shows…”
I would suggest accepting this manuscript after corrections. FPM has great potential for the future imaging technologies.

Comments on the Quality of English Language

Minor corrections are required.

Author Response

A point-by-point response is enclosed.

Reviewer 2 Report

Comments and Suggestions for Authors

the paper is focused to Fourier stack microscopy and applications, with an overwhelming emphasis on computational imaging, but some general concept of microscopy/imaging - e.g. the relation between optical resolution and Numerical Aperture - should be explained and discussed sufficiently.

There is a very little number of equations, not well interlinked to the text and paper logical flow.
The images and schematic used are many, but not easily linked in the general part of the paper rationale.

Even if a review cannot intrinsically go in to much depth, a clear and concise explanation with quantitative description of key concepts like field of view and depth of field, diffraction limit/resolution definition and numerical aperture and practical aspects of imaging systems should be given, and/or improved in the work.
The goal of the computational effort and related methodological/technical innovations could be better defined, like the rationale for overcoming the optical/practical NA limitation by stack images, etc.

Some of the references for the general part seem rather arbitrary and not fit for purpose (starting e.g. with reference 1).

Comments on the Quality of English Language

ok

Author Response

A point-by-point response is enclosed.
